# Smart Farming: Internet of Things (IoT)-Based Sustainable Agriculture

**Muthumanickam Dhanaraju [1,*], Poongodi Chenniappan [2], Kumaraperumal Ramalingam [1], Sellaperumal Pazhanivelan [3] and Ragunath Kaliaperumal [3]**

1   Department of Remote Sensing and GIS, Tamil Nadu Agricultural University, Coimbatore 641 003, Tamilnadu, India
2   Department of Electronics and Communication Engineering, Bannari Amman Institute of Technology, Sathyamangalam 638 401, Tamilnadu, India
3   Water Technology Centre, Tamil Nadu Agricultural University, Coimbatore 641 003, Tamilnadu, India
*   Correspondence: muthumanickam.d@tnau.ac.in

**Abstract:** Smart farming is a development that has emphasized information and communication technology used in machinery, equipment, and sensors in network-based hi-tech farm supervision cycles. Innovative technologies, the Internet of Things (IoT), and cloud computing are anticipated to inspire growth and initiate the use of robots and artificial intelligence in farming. Such ground-breaking deviations are unsettling current agriculture approaches, while also presenting a range of challenges. This paper investigates the tools and equipment used in applications of wireless sensors in IoT agriculture, and the anticipated challenges faced when merging technology with conventional farming activities. Furthermore, this technical knowledge is helpful to growers during crop periods from sowing to harvest; and applications in both packing and transport are also investigated.

**Keywords:** crop management; sustainable agriculture; smart farming; internet-of-things (IoT); advanced agriculture practices; issues and problems

## 1. Introduction

Sustainable agriculture is a measure of the endurance and sustenance of food grains produced in an eco-friendly manner [1]. Sustainable agriculture helps in the encouragement of farming practices and approaches to help sustain farmers and resources. It is economically feasible and maintains soil quality, reduces soil degradation, saves water resources, improves land biodiversity, and ensures a natural and healthy environment [2]. Sustainable agriculture plays a significant role in preserving natural resources, halting biodiversity loss, and reducing greenhouse gas emissions [3].

Sustainable agriculture farming is a method of preserving nature without compromising the future generation's basic needs, whilst also improving the effectiveness of farming. The basic accomplishments of smart farming in terms of sustainable agriculture are crop rotation, the control of nutrient deficiency in crops, the control of pests and diseases, recycling, and water harvesting, leading to an overall safer environment. Living organisms depend on the nature of biodiversity, and are contaminated by waste emissions, the use of fertilizers and pesticides, degraded dead plants, etc. The emission of greenhouse gases affects plants, animals, humans, and the environment; hence, it necessitates a better environment for living things [4] (Figure 1).

Agriculture is the largest contributor in India, with an 18% gross domestic product involving approximately 57% of people in rural areas. Over the years, although India's total agronomic output has increased, the number of growers has fallen from 71.9% in 1951 to 45.1% in 2011 [5]. The Economic Survey 2018 revealed that the number of agricultural workers in the total workforce will drop to 25.7% in 2050. In rural areas, farming families gradually lose the next generation of farmers, overwhelmed by higher costs of cultivation,

low per capita productivity, inadequate soil maintenance, and migrations to a non-farming or higher remunerative occupation. Presently, the world is on the verge of a digital revolution, and so it is the appropriate time to connect the agricultural landform with wireless technology to introduce and accommodate digital connectivity with farmers.

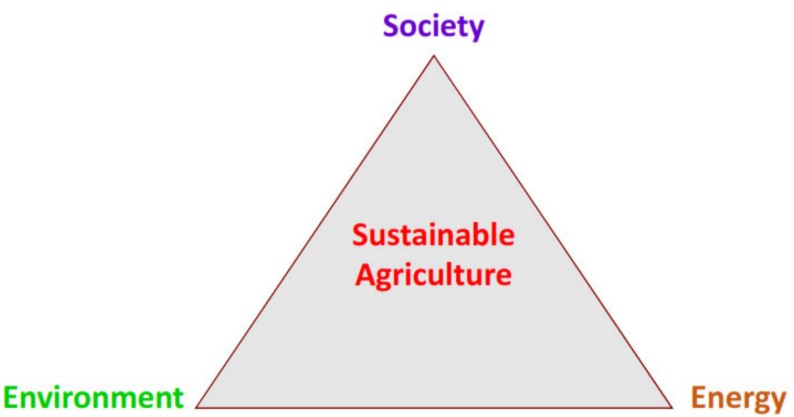

**Figure 1.** Factors of sustainable agriculture.

Regrettably, not all parts of the Earth's surface are suitable for agriculture due to various restrictions, such as: soil quality, topography, temperature, climate, and most relevant cultivable areas are also not homogenous [6]. Further, existing farming land is fragmented by political and fiscal features, and rapid urbanization, which consistently increases pressure on arable land availability (Figure 2). Recently, total agricultural land used for food production has declined [7]. Furthermore, every crop field has different critical characteristics, such as soil type, flow of irrigation, presence of nutrients, and pest resistance, which are all measured separately both in quality and quantity regarding a specific crop. Both spatial and temporal differences are necessary for optimizing crop production in the same field by crop rotation and an annual crop growth development cycle [8].

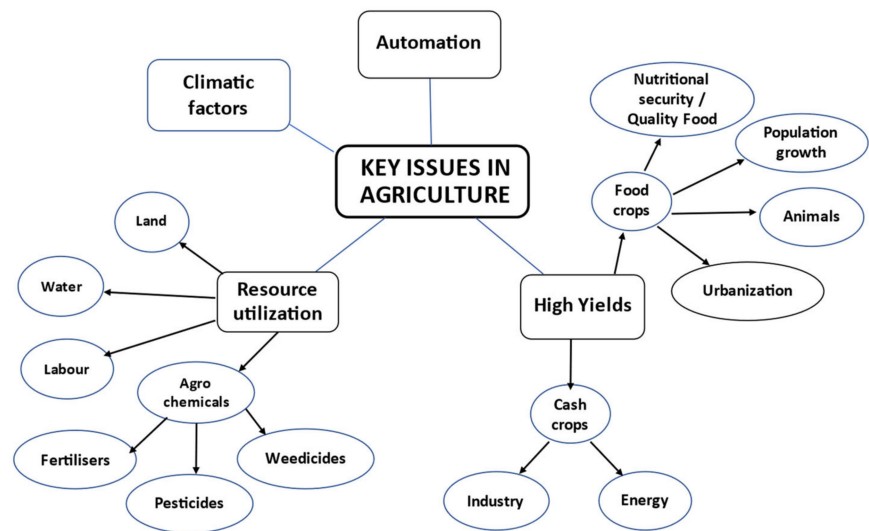

**Figure 2.** Key issues of technology in the agriculture industry.

In most cases, variations in characteristics occur within a single crop, or the same crop is grown on the whole farm and requires site-specific analyses for optimum yield production. New technology-based approaches are needed to produce more from less land, and to address these various issues. In traditional farming practices, farmers frequently

visit their fields throughout the crop's life in routine farming activities to better understand the crop conditions [9]. The current sensor and communication technologies offer an precise view of the field, from which farmers can detect ongoing field activities without being in the field in person. Wireless sensors monitor the crops with higher accuracy and detect issues at early stages, often facilitating the use of smart tools from initial sowing to the harvest of crops [10].

The timely use of sensors has made the entire farming operation smart and cost-effective, due to precise monitoring. The various autonomous harvesters, robotic weeders, and drones have sensors attached to collect data over short intervals. However, the vastness of agriculture puts extreme demands on technological solutions for sustainability with minimum ecological impact. Sensor technology through wireless communication helps farmers to know the various needs and requirements of crops without being in the fields, and they are then able to take remote action [11].

## 2. Smart Farming

Historically, ancient agriculture practices were related to the production of food in cultivated lands for the survival of humans and the breeding of animals [12], and was called the traditional agricultural era 1.0. This mainly resorted to using manpower and animals. Simple tools were used for farming activities, such as sickles and shovels. Work was mainly conducted through manual labor, and subsequently, productivity continued at a low level (Figure 3).

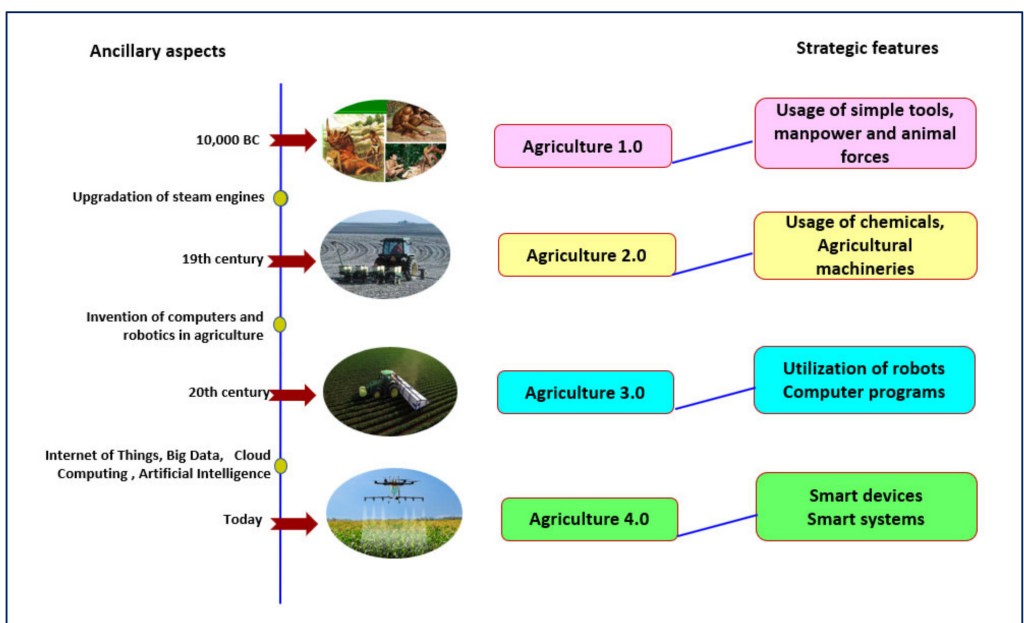

**Figure 3.** Agricultural decision support system framework.

During the 19th Century, new types of machinery appeared in the agricultural industries, in the form of steam engines. The wide use of agricultural machinery and abundant chemicals by farmers signaled the start of the agricultural era 2.0, and outwardly improved effectiveness and productivity of farmers and farms. However, considerably harmful implications, such as chemical pollution, environmental devastation, waste of natural resources, and excess utilization of energy, simultaneously developed.

The agricultural era 3.0 emerged during the 20th Century, due to the rapid growth of computation and electronics. Robotic techniques, programed agricultural machinery, and other technologies enhanced the agricultural processes efficiently. The issues that had arisen during agricultural era 2.0 were solved, and policies were readapted to the agricultural era

3.0 through work distribution, precise irrigation, the reduced use of chemicals, site-specific nutrient application, and efficient pest control technologies, etc.

The next agricultural era is also the current iteration of agriculture, the agricultural era 4.0, involving the engagement of recent technologies, such as the Internet of Things, big data analysis, artificial intelligence, cloud computing and remote sensing, etc. The adoption of new technologies has significantly improved agricultural activities by developing low-cost sensor and network platforms, aimed towards the optimization of production efficiency, along with reductions in the usage of water resources and energy with minimum environmental effects [13]. Big data in smart farming provides extrapolative overviews of real-time agricultural situations, allowing farmers to make effective decisions [14]. Real-time programming is developed with artificial intelligence concepts and embedded in IoT devices, helping farmers make the most suitable decisions [15].

Smart farming promotes precision agriculture with modern, sophisticated technology and enables farmers to remotely monitor the plants. Smart farming helps agricultural processes, such as harvesting and crop yields, as the automation of sensors and machinery has made the farming workforce more efficient [16]. The technologies convert traditional farming methods to automatic devices, causing a technological revolution in agriculture. Today, the technology in agriculture has altered the way farming is conducted, and conventional techniques have been transformed by the Internet of Things [17].

In terms of optimizing farm labor requirements and increasing the quantity and quality of products, smart farming is an emerging modern technique implemented with information and communication technologies (ICT) [16]. Modern ICT technologies, such as the Internet of Things, GPS (Global Positioning Systems), sensors, robotics, drones, precision equipment, actuators, and data analytics, are used to identify the farmers' needs and select suitable solutions to their problems. These innovations increase the accuracy and timeliness of decisions taken, and improve crop productivity. Several multilateral organizations and developing countries around the world have proposed smart farming technologies to increase agricultural output [18].

Sensors are constantly monitoring crops with greater accuracy, detecting any undesirable conditions during the early stages of the crop's lifecycle. Current farming incorporates smart tools from crop sowing to harvest, storing, and conveyance. The appropriate use of a wide variety of sensors has made the entire operation both more efficient and profitable, due to its accurate monitoring competencies. In addition, sensors that collect data quickly are directly available online for further evaluation, and they provide crop and site-specific agriculture for every site.

The many issues related to crop production are addressed by smart agriculture and monitoring, particularly regarding changes in soil characteristics, climate factors, soil moisture, etc., to improve the spatial management practices that increase crop production and avoid the excess use of fertilizers and pesticides [19]. The ANN models in smart irrigation water management (SIWM) regulate irrigation scheduling support systems (DSS) and offer data on irrigation efficiency, water productivity index, and irrigation water demand and supply on a real-time basis. Climate-smart agriculture (CSA) is an upcoming technology, especially in developing countries, due to its potential to improve food security, farm system resilience, and lower greenhouse gas emissions [20]. Smart agriculture technology based on IoT technologies has many advantages in all agricultural processes and practices in real-time, including irrigation, plant protection, improving product quality, fertilization, disease prediction, etc. [21]. The benefit of smart agriculture lies in its collection of real-time data on crops, the precise assessment of soil and crops, remote monitoring by farmers, supervising water and other natural resources, and improving livestock and agricultural production. Therefore, smart agriculture is considered to be the progression of precision agriculture through modernization and smart methods to attain various information of farm activities that are then remotely managed, and reinforced by suitable alternative real-time farm maintenance solutions.

### 3. Internet of Things

The Internet of Things (IoT) is a new technology that allows devices to connect remotely to achieve smart farming [22]. The IoT has begun to influence a vast range of industries, from health, trade, communications, energy and agriculture, to enhance efficiency and performance across all markets [23–25].

Current applications provide information on the IoT's effects, and its practices that are yet to be observed. However, by considering the advancement of technologies, one can envisage the IoT technologies perform a crucial role in numerous activities of farming, such as the utilization of communication infrastructure, data acquisition, smart objects, sensors, mobile devices, cloud-based intelligent information, decision-making, and the automation of agricultural operations (Figure 4).

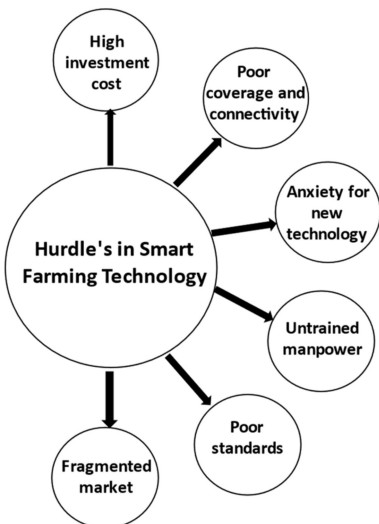

**Figure 4.** Barriers in the implementation of smart agriculture technology.

The IoT technology monitors plants and animals and remotely retrieves information from mobile phones and devices. Sensors and instruments empower farmers to assess the weather and to anticipate production levels. The IoT plays a role in water harvesting, monitoring and controlling the flow amount, assessing crops' water requirements, time of supply, and the saving of water, more than ever before [26]. Sensors and cloud connectivity through the gateway can remotely monitor the status and water supply based on soil and plant needs [27]. To correct nutrient deficiencies, pests, and diseases, farmers cannot monitor and observe every plant manually, but IoT technology is still beneficial and has led farmers to a new milestone in modern agriculture [28].

Recently, the development of IoT technologies has played a major role throughout the farming sector, particularly through its communication infrastructure. This has included connecting smart objects, remote data acquisition, using vehicles and sensors through mobile devices and the internet, cloud-based intelligent analysis, interfacing, decision formation, and the automation of agricultural operations. These proficiencies have revolutionized the agriculture industry in terms of resource optimization, controlling climate effects, and improving crop yields.

Researchers have proposed different methods, architectures, and various equipment to monitor and convey crop information at different growth stages, based on several crop and field types. Many manufacturers provide communication devices, multiple sensors, robots, heavy machinery, and drones to collect and then distribute data. Food and agriculture organizations, along with other government organizations, develop guidelines and policies for regulating the use of technologies to preserve food and environment safety [29,30].

*Fundamentals of IoT Applications in Agriculture*

The accessible, inexpensive and interactive tracking platform provides consolidated information on traditional agricultural methods, techniques, implements, crop pests and diseases, etc., collected from various sources for sustainable agriculture. Interactive agriculture allows easy access to the data by users through multiple devices, such as computers and mobile phones [31].

1. Robust Models: The distinctive features of the agriculture sector are diversity, complexity, spatio-temporal variability, and uncertainties of the right types of harvests and facilities.
2. Scalability: The variation in farm size from smaller to larger; hence, the results should be scalable. The placement and testing planning should be progressively scaled up with fewer expenses.
3. Affordability: Affordability is vital to farming achievement, and therefore price should be suitable with significant assistance. Standardized platforms, products, tools, and facilities could obtain a satisfactory price.
4. Sustainability: The problem of sustainability is a vital issue due to strong economic pressure and intense competition worldwide.

## 4. Technologies Used in Smart Farming

### 4.1. Global Positioning System (GPS)

GPS accurately records latitude, longitude, and elevation information [32]. Global Positioning System satellites transmit signals and permit GPS receivers to compute their location in real-time, and provide continuous positions while moving. The exact location information offers farmers the opportunity to discover the precise position of field data, such as pest occurrence, type of soil, weeds, and other barriers. The system facilitates the recognition of various field locations in order to then apply the necessary inputs (seed, fertilizer, herbicide, pesticide, and water) to a particular field [33].

### 4.2. Sensor Technologies

Technics, such as photo electricity, electromagnetics, conductivity, and ultrasound, are used to estimate soil texture and structure, nutrient level, vegetation, humidity, vapor, air, temperature, etc. Remote sensing data can differentiate between crop types, categorize pests and weeds, locate stress in soil and plant conditions, and monitor drought [34].

Plant health is affected by many factors, such as soil moisture, nutrient availability, exposure to light, humidity, the amount of rainfall, the color of leaves, etc. The plants are monitored by maintaining the optimum temperature and light intensity, and conserving water and energy through micro-irrigation. Different sensors are used to detect many parameters. If they cross a threshold, the sensor senses the changes and transmits them to the microcontroller to perform the required actions until the parameter returns to its optimum level [1].

The temperature, humidity, soil pattern monitoring, airflow sensor, location, $CO_2$, pressure, light, and moisture sensors are generally used in sensing technologies. Prominent sensor characteristics include reliability, memory, portability, durability, coverage, and computational efficiency, and make them suitable for agriculture [35]. Currently available wireless sensors play a vital role in collecting data on crop conditions and providing other information. These sensors are standalone types and can be integrated with advanced agricultural tools and heavy machinery, based on application necessities.

The major sensor types with their corresponding working procedure and purpose are represented in Table 1.

**Table 1.** Sensor types and their applications.

| Sensors | Applications | Working Procedure |
|---|---|---|
| Acoustic sensors | Pest monitoring and detection classifying seed varieties, fruit harvesting [36]. | Measuring the variations in noise level when intermingling with other materials, i.e., soil particles [37]. |
| Airflow sensors | Measuring soil air permeability, moisture, and structure in a static position or mobile mode [38]. | Based on various soil properties, unique identifying signatures [38]. |
| Eddy covariance-based sensors | Quantifying exchanges of $CO_2$, water vapor, methane, or other gases. Measuring surface atmosphere and trace gas fluxes in various agricultural ecosystems [39]. | Measuring continuous flux over large areas [40]. |
| Electrochemical sensors | To analyze soil nutrient levels and pH [41]. | Nutrients in soil, salinity, and pH are measured using sensors [42] |
| Electromagnetic sensors | Recording electrical conductivity, electromagnetic responses, residual nitrates, and organic matter in soil [43]. | Electrical circuits measure the capability of soil particles to conduct or accumulate electrical charge [44]. |
| Field programmable gate array (FAAA) based sensors | Measuring real-time plant transpiration, irrigation, and humidity [45]. | Programmable silicon chips and logic blocks are surrounded together by programmable interconnected resources of the digital circuit [46]. |
| Light detection and ranging (LIDAR) | Land mapping, soil type determination, farm 3D modelling, erosion monitoring and soil loss, and yield forecasting [47]. | Sensors emit pulsed light waves and bounce off when colliding with objects and are returned to the sensor. The time taken for each pulse to return is used for assessment [47]. |
| Mass flow sensors | Yield monitoring based on the amount of grain flow through a combine harvester [48]. | Sensing the mass flow of grain with modules, e.g., grain moisture sensor, data storage device, and an internal software [48] |
| Mechanical sensors | Soil compaction or mechanical resistance | Sensors record the force assessed by strain gauges or load cells [48]. |
| Optical sensors | Soil organic substances, soil moisture, color, minerals, composition, clay content, etc. Fluorescence-based optical sensors are used to supervise fruit maturation [49]. Integrating optical sensors with microwave scattering to characterize orchard canopies [50] | Sensors use light reflectance phenomena to measure changes in wave reflections [44]. |
| Optoelectronic sensors | Differentiate plant types to detect weeds in wide-row crops [51]. | Sensors differentiate based on reflection spectra [51]. |
| Soft water level-based (SWLB) sensors | Used in catchments to characterize hydrological behaviors (water level and flow, time-step acquisitions) [52] | Measuring rainfall, stream flow, and other water presence options [52]. |
| Telematics sensors | Assessing location, travel routes, and machine and farm operation activities [53]. | Telecommunication between places (especially inaccessible points) [53]. |
| Ultrasonic ranging sensors | Tank monitoring, spray distance measurement, uniform spray coverage, object detection, monitoring crop canopy [54], and weed detection [55]. | An ultrasonic sensor uses a transducer to send and receive ultrasonic pulses that relay information about an object's proximity [56]. |
| Remote sensing | Crop assessment, yield modeling, forecasting yield date, land cover and degradation mapping, forecasting, the identification of plants and pests, etc. [57]. | Satellite-based sensor systems collect, process, and disseminate environmental data from fixed and mobile platforms [57]. |

### 4.3. Variable-Rate of Technology (VRT) and Grid Soil Sampling

Variable-rate technologies (VRT) are used in farming to predict the delivery rate of inputs based on a predetermined map extrapolated from GIS for the placement of inputs at variable amounts in the right place and at the right time [16,33]. Grid soil sampling is soil collection from a systematic grid to establish a map for every parameter. These maps are the basis for VRT and are loaded into a variable-rate applicator. The computer and GPS receiver direct and control the changes in the delivery amount or fertilizer product, based on map features [58,59].

New technologies, such as variable rate technology and associated practices (grid soil sampling), potentially improve soil fertility management and assess the spatial distribution of nutrients and yields [60]. In grid sampling, samples are collected from sampled sections based on the subdivision of a field into small areas, or cells, by superimposing the grid lines onto the field. Composite samples represent an entire area of each much smaller area (grid-point sampling) at the intersections of grid lines. Soil-test values from grid sampling are mapped by interpolating methods from non-measured locations between sampled points. The variability of phosphorus and potassium is field-specific, and each field should be fertilized differently to improve nutrient management practices by uniform applications of fertilizers and manure for better precision agriculture [61].

### 4.4. Geographic Information System (GIS)

The GIS comprises hardware and software designed to provide compilation, storage, retrieval, attributes analysis, and location data to generate maps and analyze characters and geography for statistics and spatial methods [62]. The GIS database provides information on field soil types, nutrient status, topography, irrigation, surface and subsurface drainage, quantity of chemical applications, and crop production, and also establishes the relationship between elements that affect a crop on a particular farming field [63]. Apart from data storage and display, the GIS is used to assess present and alternative management by compounding and altering data layers for decision-making.

### 4.5. Crop Management

Satellite images provide information on variations in soil conditions, as well as crop performances affected by topography within the field. Therefore, farmers can exactly monitor production factors, such as seeds, fertilizers, and pesticides, that are responsible for yield increase and efficiency.

The spatial coverage and temporal revisit frequency of satellite images provide the information in near real-time at a regional scale. The relationship between the spectral properties of crops and their biomass/yield experiments [64] is predicted by spectral reflectance properties of vegetation, especially in red and near-infrared combinations (vegetation indices) to monitor green foliage. Among the different indices, the normalized difference vegetation index (NDVI) is the most popular indicator to assess vegetation health and crop production, due to the closely related leaf area index (LAI) and photosynthetic activity of green vegetation [25]. Crop monitoring methods are based on the interpretation of remote-sensing-derived indicators by comparing actual crop status to previous or normal seasons [65]. The relationship between vegetation indices and biomass permits early crop yield estimation in certain periods before harvest [66]. The automated data acquisition, processing, monitoring, decision-making, and management of farm operations [67], including the basic functions of crop production (yields), profits and losses, farm weather prediction, field mapping, soil nutrients tracking, are the more complicated functionalities available through automated field management.

### 4.6. Soil and Plant Sensors

Sensor technology, a significant constituent of precision agriculture, provides soil properties information, fertility, and water status. Hence, new sensors have been developed based on desirable features and established apart from currently available sensors [68].

Soil sensors and plant wearables monitor real-time physical and chemical signals in soil, such as moisture, pH, temperature, and pollutants, and provide information to optimize crop growth conditions, fight against biotic and abiotic stresses, and increase crop yields. Soil organic matters (SOMs), nitrogen (N), phosphorus (P), and potassium (K) are the most important nutrients for crop production. The NIR reflectance-based sensors measure the spatial variation of surface and subsurface soil nitrogen [69]. SOM is predicted based on optimal wavelengths by assessing soil spectral reflectance in IR and visible wavelength regions [70]. The soil nitrogen and phosphorus are predicted using NIR spectrophotometry technology [71–73]. The soil apparent electrical conductivity (ECa) sensors collect information continuously on the field surface, since ECa is sensitive to changes in soil texture and salinity. Soil insects/pests are detected using optoelectronic, acoustic, impedance sensors, and nanostructured biosensors [74].

### 4.7. Rate Controllers

Rate controllers are designed to control the delivery rate of inputs by monitoring the speed of vehicles across the field, and altering the flow rate of material on a real-time basis at the target rate. Rate controllers are commonly used as stand-alone systems [75].

### 4.8. Precision Irrigation in Pressurized Systems

Recent developments in irrigation systems have introduced irrigation machines, devoted to motion control, GPS-based controllers, sensor technologies, and wireless communication to monitor soil and climatic conditions together with an assessment of irrigation parameters, i.e., flow and pressure, to attain greater water utilization efficiency by crop. These technologies show significant potential; however, further progress is required before they can become commercially available [76].

### 4.9. Yield Monitor

Yield monitors are the combination of sensors and components, including a data storage device, a computer, and user interface, that control integration and interaction components. The sensor measures yield continuously by evaluating the force of mass or volume of grain flow. The mass flow sensor was based on the principle of transmitting microwave energy beams and measuring the energy that bounces back after hitting. In yield monitors, GPS receivers create yield maps based on the location yield data [77].

The yield monitor is mounted on a harvester and connected with the mobile app for displaying live harvest data, and automatically uploads to the web-based platform. The app can generate and share high-quality yield maps with an agronomist, and farmers can export other farm management data for analysis. In horticultural crops, to precisely determine the yield quantity and quality of produce, fruit growth is considered one of the most relevant parameters in the crop progressing period [78]. Color images are used to track fruit conditions for estimating fruit maturation, making decisions for harvesting, and targeting the right market [79]. Satellite images are one of the options for real-time monitoring of the yield of crops over vast areas; for example, Sentinel-1A images are used to map the rice yield and crop intensity in Myanmar [80].

The crop yield estimation system was designed using both software and hardware components. Based on a Bluetooth terminal android application and yield estimator software program, crop yield is estimated using a mathematical calculation through a mobile application [81]. Satellite-based crop yield predictions based on spectral signatures reveal the estimated yields are as reliable as actual yields. The maize yield predictions were successfully carried out under varying environments using machine learning and satellite-derived data assimilation in crop models [82].

### 4.10. Software

The software has multiple tasks, such as mapping, display controller interfacing, data processing, analysis, and interpretation, etc. Most commonly, software is used to generate

the maps for soil properties and nutrient status, yield maps, variable rate applications maps for inputs, and overlaying different kinds of maps with advanced geostatistical features [83].

## 5. Applications in Agriculture

By adopting the current sensor and IoT technologies in agriculture, each characteristic of conventional farming practices is rehabilitated. The incorporation of wireless sensors and IoT in smart farming answers many of the issues facing conventional agriculture; for example, land suitability, drought monitoring, irrigation, pest control, and yield maximization. Figure 5 demonstrates the order of main applications, facilities, and devices for smart agriculture applications. Using advanced technologies at various stages in the following few applications enhances efficiency and revolutionizes agriculture.

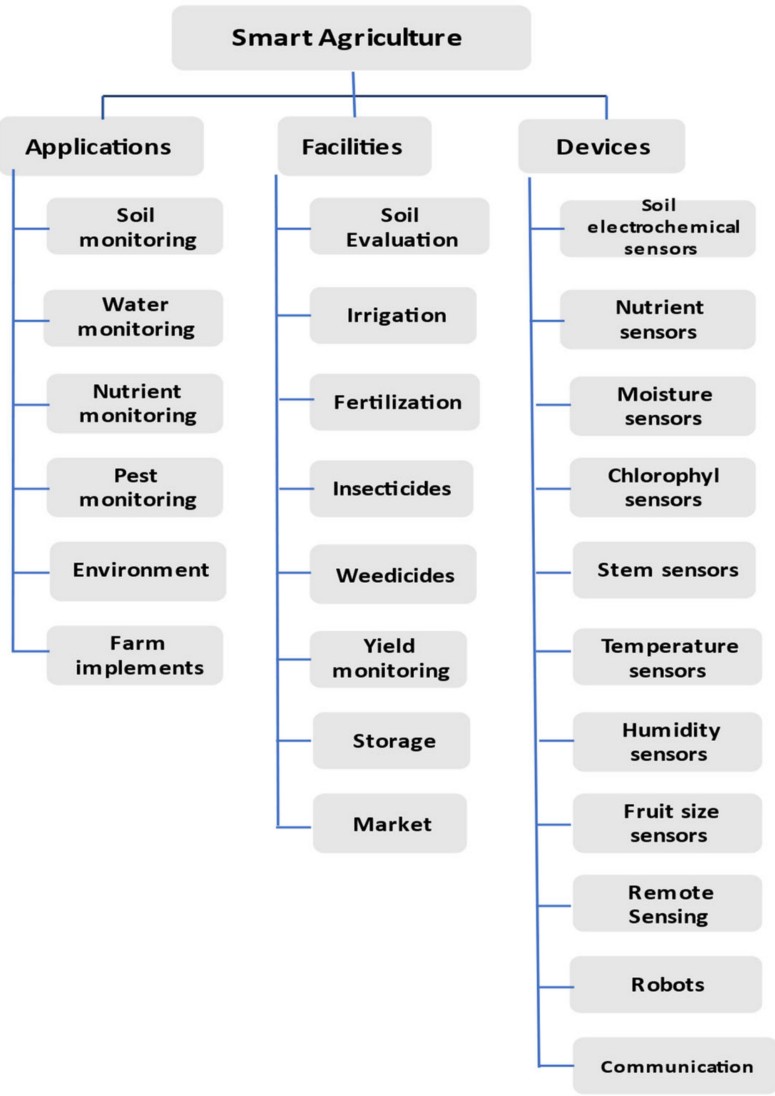

**Figure 5.** Hierarchy of probable applications, facilities and devices for smart agriculture.

### 5.1. Soil Mapping and Plant Monitoring

Soil analysis estimates the nutrient status of the field based on GPS position and field-specific information, and critical decisions are then taken according to the nutrient deficiencies at different stages of the crop. The factors controlling soil fertility status are topography, type and texture, cropping pattern, application of fertilizer, irrigation, etc. [84]. Soil mapping is useful for assessing crop suitability or varieties in a specific field, as well as planting depth, the physical, chemical and biological properties of soil, in order to best utilize resources. Presently, a wide range of sensors and tools are used to monitor soil properties, such as water-holding capacity, texture, and absorption rate, which assists farmers in tracking the soil quality and adopting suitable remedies to avoid soil degradation such as erosion, alkalization, acidification, salinization, and pollution. Drought is another concern that affects plant productivity and crop yield. Remote sensing techniques that can obtain soil moisture data frequently assist in analyzing agricultural drought in remote regions. Soil moisture maps generated from satellite data are used to estimate the soil water deficit index (SWDI), which enables the development of prediction models based on soil physical properties [85,86].

Various factors, such as soil type, soil nutrients, irrigation, and pests, affect rice yield and quality. The IoT-based mobile application aids crop management and provides real-time information on soil nutrition and characteristics. The system consists of electrical conductivity (EC), temperature sensors with a T-Beam microcontroller, and IoT connectivity, and the estimated EC value near the calibration solution is 12.88 mS/cm, and 150 mS/cm is less than 2% of the calibration solution's value. The measured EC values are linearly proportional to temperature and depth, and values of 1.04 and 3.86 mS/cm were noticed with and without fertilizer at 5 cm depth, while it was 0.656 and 420 mS/cm at 10 cm depth, respectively [87].

Plant monitoring conducted through the IoT ADCON-based station, with sensors and mobile devices (smartphones and tablets), farmers are able to collect data on soil and ambient parameters, such as leaf wetness, air and soil temperature, soil and air humidity to improve the grape productivity, and crop quality from seeding to harvest. Further, the data transmission system highlights the soil-plant-atmosphere interactions needed to optimize agricultural production [88]. By analyzing the data from soil moisture, carbon dioxide, light, and temperature sensors in bell peppers grown in a greenhouse were compared with day and night $CO_2$, rolling the doors and windows of the greenhouse open and closed, based on soil moisture [89].

### 5.2. Irrigation

According to the UN Convention to Combat Desertification (UNCCD), 168 countries will be inundated with desertification by 2030, and nearly 50% of the world population lives in high water shortage areas [90]. Considering the water crises and increasing demand for farming and other activities, it must be provided to regions with water quantities. Water resources are conserved by adopting more controlled and efficient irrigation systems; for example, drip and sprinkler irrigations. Water demand estimation for crops is controlled by soil type, precipitation, irrigation method, crop type, and requirement, as well as soil moisture retention. Using air and soil moisture control systems with wireless sensors optimizes water resources and improves crop health. In the current scenario, a substantial increase in crop productivity is anticipated using IoT techniques, namely CWSI (crop water stress index)-based water management [91], calculated from the crop canopy at varying crop growth stages and air temperatures. The information from climate data, sensors, and satellite imaging are related to the CWSI model for water requirement calculation, and predictions using the irrigation index values can be used for every field based on slope or soil variability to improve water usage efficiency.

### 5.3. Site-Specific Nutrient Management

Fertilizer is a either natural or synthetic chemical substance that provides nutrients for plant growth and soil fertility. Both nutrient deficiency and excessive fertilizer use harms soil, plant health, and the environment [92]. The site-specific soil nutrient fertilization under smart agriculture estimates the required quantity of nutrients precisely, and minimizes their negative effects through excessive use on soil and in the environment. The site-specific soil, nutrient measurements are influenced by soil types, crop type, yield targets, exchange capacity, use efficiency, the type of fertilizer, weather conditions, etc. The IoT-based fertilizing technique estimates the nutrient's spatial patterns of distribution [93,94]. The normalized difference vegetation index (NDVI) was obtained from satellite images to observe crop nutrient status [95,96], crop health, vegetation vigor, and plant density, as well as soil nutrient level. Recent technologies, like GPS [97], geo mapping [98], variable rate technology (VRT) [99,100], and autonomous vehicles [101] strongly contribute to IoT-based smart fertilization. Apart from these, fertigation [102] and chemigation [103,104], i.e., the use of water-soluble fertilizers in soil amendments and pesticides, are considered effective management practices to improve fertilization efficiency.

### 5.4. Crop Pest and Disease Management

The Food and Agriculture Organization (FAO) concluded that an annual global crop yield loss of 20- 40% was only due to pests and diseases [105], and these losses are controlled by the use of pesticides and other agrochemicals [106]. Most of them are harmful to human and animal health, and ultimately cause contamination of environmental systems [107,108]. The IoT-based devices, such as robots, wireless sensors, and drones, precisely spot and control the crop opponents by real-time monitoring, modelling, and disease forecasting, increasing overall effectiveness [109,110] more than traditional pest control procedures. The IoT-based disease and pest management process depends on detection and image processing. The remote sensing imagery and field sensors are used to collect data, such as plant health and pest incidence, in every field for the entire crop period. IoT-based automated traps [111,112] capture, count, and describe insect types, and further upload data to the Cloud for complete analysis. Due to advancements in robotic technology, an agricultural robot with multispectral image sensing devices and precision spraying nozzles is utilized to detect and control pest problems more accurately under the IoT management system.

### 5.5. Yield Monitoring and Forecasting

The yield monitoring mechanism conforms to yield, moisture content, and quality of produce. The quality depends on pollination with good pollen, especially under changing environmental circumstances [113–115]. Crop forecasting predicts the yield before the crop harvest, and assists the farmer in future planning, decision-making, and further analysis of the yield quality. Maturity determines the right harvesting time by monitoring the crop at different development stages, including factors such as fruit color, size, etc. Predictions of the correct harvesting time aids in maximizing crop quality and production, and regulates market management strategies. Therefore, farmers should know the exact harvest time of crops to obtain profit. Figure 6 outlines the idea of a farm area network, representing the whole farm in real-time conditions.

The development and installation of a yield monitor [116] on a harvester, connected with a mobile app, shows real-time crop harvest, and automatically transmits data to the manufacturer's web-based platform. To estimate crop production and monitoring, satellite images are exploited to cover vast areas [80]. For fruit crops, multicolor (RGB) satellite images [79] are utilized to track the diverse fruit conditions, especially fruit size and color, and plays a major role in estimating its maturation, making decisions on harvest, and market opportunities. Similarly, multiple optical sensors are used [117] to monitor shrinking fruits during drying conditions.

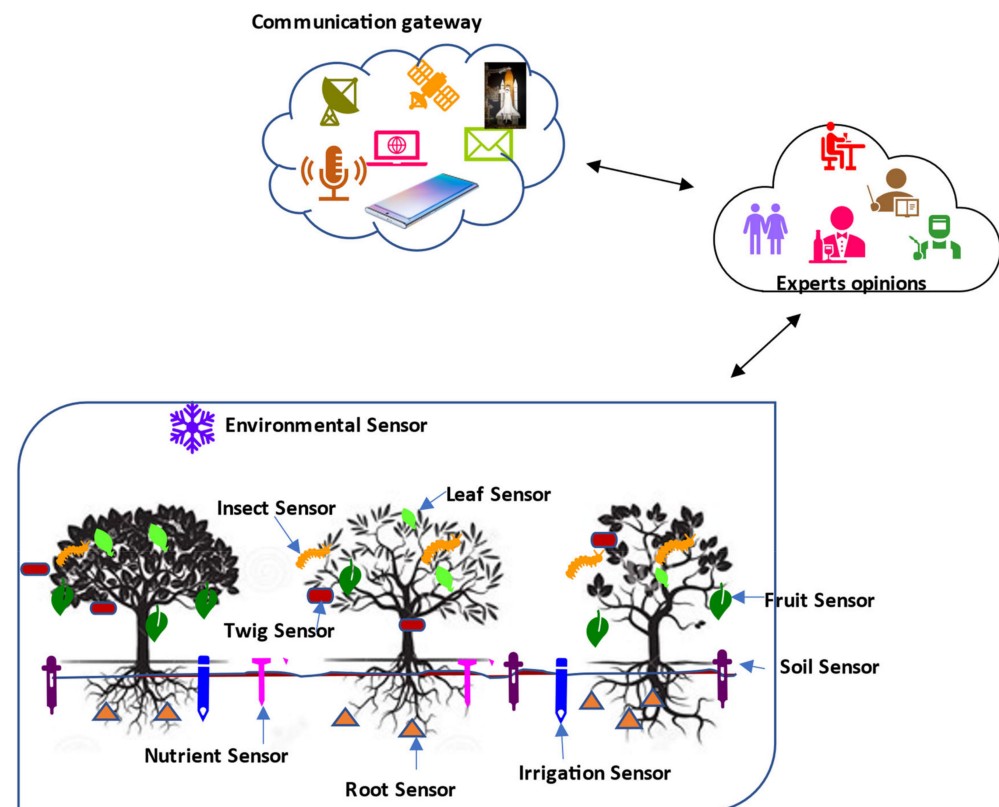

**Figure 6.** An Internet-of-Things-based network for smart farming.

## 6. Role of IoT in Advanced Farming Practices

Adopting the new methods based on sensor and IoT-based technologies improved the yield of crops more than conventional agriculture processes. The involvement of new sophisticated sensor-based technologies in controlled environments plays an important role in enhancing the quality and quantity of produce.

### 6.1. Greenhouse Farming and Protected Cultivation

Growing plants in a controlled environment gained popularity in the 19th Century, and is considered one of the oldest methods of smart farming. These practices further accelerated during the 20th Century in countries facing severe weather conditions [118]. Crops grown in indoor conditions are less affected by the environment. As a result, crops grown traditionally under suitable conditions are today being raised at anytime and anywhere by the use of sensors and communication devices. The success of crop production under a controlled environment depends on various factors, such as shed structures and material for controlling wind effects, aeration systems, accuracy of monitoring parameters, decision support system, etc. [119]. One of the greatest challenges in greenhouses is the precise monitoring of environmental parameters; hence, it requires several measurement points to predict the various parameters for controlling and ensuring the local climate. In an IoT-based greenhouse, sensors are used to measure and monitor the internal parameters, such as humidity, temperature, light, and pressure [120].

The smart greenhouse has helped farmers automatically conduct farm work, without manual inspection, and protects the plants from hailstorms, winds, ultraviolet radiation, and insect and pest attacks. Hibiscus plants are grown with the required wavelength during the night using lights, temperature, and air humidity sensors. A study revealed a reduction in 70–80% water requirement, and the IoT enables direct contact between the farmer and consumer to make farming as efficient and profitable as possible [121]. The IoT-enabled automated system increased the productivity of rose plants grown in a greenhouse by

monitoring and controlling various parameters, such as humidity, mist, $CO_2$ level, UV light intensity, pH and EC value, water nutrients solution level, temperature, and amount of pesticides, through sensors for further efficient detection and diagnosis [122].

*6.2. Hydroponics*

Hydroponics, a subdivision of hydroculture, is growing plants without soil to improve greenhouse farming benefits. Hydroponic-based irrigation systems enable a balanced rate of application of dissolved nutrients in the water to crop roots as a solution. Presently, the available systems and sensors [123] detect a wide range of parameters and perform data analysis at predetermined intervals. Precise measurement and monitoring of nutrient content in solution is crucial for plant growth and considers its demands. On a real-time basis, the wireless-sensor-based prototype [124] has delivered a solution for soilless cultivation, and measures the concentration of numerous nutrients and water levels [125].

An automated smart hydroponics system integrated with IoT consists of three major components: input data, cloud server and output data. These monitor lettuce cultivation from anywhere through the internet by analyzing parameters, such as pH level, water, nutrient-rich water-based solution, room temperature, and humidity, on a real-time basis [126]. The hydroponic system of the deep flow technique is a method for cultivating plants by placing roots in deep water layers, and ensuring the continuous circulation of plant nutrient solution. The plant growth elements data, such as pH, temperature, humidity, and water level in the hydroponic reservoir, are acquired by sensors integrated into Raspberry Pi, and data are processed and monitored automatically on a real-time basis to ensure proper water circulation [127].

*6.3. Vertical Farming*

The industrial-based agricultural farming practices damage soil quality at a faster rate than nature can reconstruct. The alarming erosion rate and use of fresh water for agriculture has led to the reduction of arable land, and increased the overburden on present water reservoirs [128]. Vertical farming (VF) offers an opportunity to keep the plants in a precisely controlled environment, significantly reducing resource consumption and, at the same time, increasing production at varied times; and only a portion of the ground surface is needed depending on the number of stacks. VF is also extremely effective in higher yields and reducing water consumption compared to traditional farming [129]. The carbon dioxide measurement is the most critical parameter; hence, nondispersive infrared (NDIR) $CO_2$ sensors play a vital part in tracking and controlling the conditions in vertical farms.

*6.4. Phenotyping*

Phenotyping is an emerging crop engineering technique, relating plant genomics with ecophysiology and agronomy. The advancement of genetic and molecular tools is significant for crop breeding; however, quantitative analysis of crop behaviors, such as pathogen resistance, grain weight, etc., is inadequate due to the absence of effective technologies and efficient techniques. In this condition, [130] reported that plant phenotyping is highly useful in investigating the quantitative characteristics responsible for growth, resistance to various stresses, yield quality, and quantity. The sensing technologies and image-based phenotyping describe screening of biostimulants and an understanding of their mode of action [131]. IoT-based phenotyping is intended to observe the crop and related trait measurements, and offer facilities for the breeding of crops and digital agriculture [132]. The trait analysis algorithms and modelling support determine the relationships among genotypes, phenotypes, and their growing condition.

**7. The Role of the Engineer in Smart Farming**

Farmers face many issues when they adopt IoT-based agriculture. Therefore, engineers must develop solutions for specific problems related to smart farming techniques. An engineering role concerns the application and use of innovative technologies and methods

for precision agricultural machinery, and smart farming is a creative way to mechanize agricultural engineering through means different from conventional mechanization [133]. The concepts and synergy-based information are obtained from different technology areas, such as agricultural mechanization, mechatronics, instrumentation, control systems, and knowledge in artificial and computational intelligence [134]. Big data, satellite, and aerial images have revolutionized precision agriculture, and these new technologies increase production efficiency by creating a balance between productivity and environmental protection. As a system integrator, engineering combines technical experience and strong business skills in both the public and private sectors [135].

At the same time, engineering exploits the rewards of digital transformation in the entire agri-food chain, from day-to-day farming activities to supporting sales operations, logistics, and the maintenance of farm assets. For example, knowledge of the IoT, AI, mobile, precision farming technologies, remote sensing, advanced analytics, the Cloud, RPA, and blockchain technologies are necessary [136]. The data collected from the various types of machinery through sensors and other devices generates responses concerning cereals, viticulture, fruit, and vegetables, as well as soil and monitoring [137].

The use of digital technologies and control systems to automate production processes also reduces manual human intervention. The production process, from field to final product, is carried out by planning, organizing, and analyzing data received from machines. The data acquired are stored in historical archives and correlated with each other to retrieve useful information for products through traceable systems working based on radio-frequency signals [138].

### 7.1. Purpose

Purpose is based on the user's final requirement, and influences the monitoring of crops during the growth period. Sensors provide the IoT solutions to their problems. For example, the end-user is a corn farmer, faced by problems mainly concerning water usage and ensuring that a crop gets adequate water; therefore, water level and moisture monitoring sensors are accommodated to prevent water wastage.

### 7.2. Technology

Distance plays an important role in technology selection because the sensors collect data and send to the server; hence, similar technology cannot be used for varying distances. For example, radio frequency identification (RFID) or near field communication (NFC) and low power, wide area network (LPWAN) technologies could send data over a distance of hundreds or even thousands of meters.

### 7.3. Power Requirements

Most IoT solutions are spread across a large farm, so it is better to develop low-power applications. On the other hand, more data transmission requires huge data costs and power consumption; hence, designers need to consider developing cost-effective IoT solutions for farmers. Usually, engineers save costs with customized IoT-based farming solutions, and develop apps for sending the data less frequently.

### 7.4. Data Frequency

The end user's necessities are critical in deciding the number of sensors and data packets. Sometimes, a farmer does not require information frequently, but developer design an IoT application to function on a continual and real-time basis, with very high data frequency.

### 7.5. Placement of Sensors

Sensors are placed in such a way that they provide optimal performance, even if the farm has all the essential sensors with proper placement.

## 8. Barriers to Implementing Smart Farming Technologies

Technology adoption is a method with a certain level of heterogeneity factors that are affective [139]. Technology implemented in farming systems has provided accuracy, efficiency, and eased time pressures. Although smart farming increases the productivity of crops, there are still problems in adopting these technologies

### 8.1. Cost of Technology

Existing technologies minimize the workforce and perform extremely fast with high accuracy. Therefore, it is anticipated that machines would probably replace a human workforce in the near future. However, it is impossible, since many countries have experienced poverty wherever the workforce was the main source for the agriculture sector. The implementation of devices and technologies requires a huge amount of money; therefore, farmers face difficulties in terms of affordability when they look beyond conventional tools.

### 8.2. Lack of Financial Resources

Financial supporters could provide adequate loans to farmers if farmers did not get the anticipated yield, perhaps because unexpected calamities like drought, flood, pests, and diseases impacted the crops.

### 8.3. Literacy Status of Farmers

The education level among the farmers is one of the greater challenges in implementing technologies in developing countries. The knowledge needed encompasses educational and technical abilities to manage the tools. The level of education increases a farmers' aptitude to process information, and thus make decisions using smart farming technologies [140], facilitating farmers' use of computers [141]. Farmers in developing nations are mostly uneducated and unskilled because of a lack of desire to gain knowledge, or any new technology awareness [142]. Hence, it is a reason for farmers in choosing traditional farming over smart farming [143]. Farmers have considered that usage is too complex, sometimes incapable of recognizing the icons used in a mobile application as the farmers use general icons based on traditional understanding. Farmers need to be digitally literate to reinforce the advantages of smart farming technologies and, simultaneously, agri-tech companies should ensure farmers easily understand the limitations of the technology.

### 8.4. Lack of Integration between the Systems

Integration across systems is one of the areas where smart farming technologies needs to be advanced further by incorporating production, property management, and decision-making tools. The communication between academics and interdisciplinary groups must overcome the gap between agricultural and information science. More emphasis has been given to increasing user effectiveness during the development of an information system [144]. The basis for improved decision-making is based on the timely obtainability of superior quality data; hence, data must be integrated to generate information and knowledge.

### 8.5. Telecommunications Infrastructure

Farming activities mostly occur in rural areas more effectively in arable land than contaminated land. However, poor telecommunication infrastructure makes data transmission unreliable, especially through mobile phones and tablets. Smart farming necessitates a real-time connection with the internet to enable the use of information. In addition, various operation control systems, such as fertilizers, pesticides, and seed volume, requires high-quality internet connection to produce outcomes. Recently, with the expansion of mobile phones, rural producers have gained to access mobile internet; however, signal quality and input speed are limited.

### 8.6. Data Management

Farmers are facing problems in organizing and manipulating data obtained by the sensors. The weather stations are generating data; however, farmers do not recognize how to use the information and how to change the data into a more available form. Its complex systems, alongside issues of acceptability and usability, lead to incorrect calculations. Farmers, consultants, and others involved in the production process must provide greater accessibility to data and information in productive systems.

## 9. Current Challenges and Future Expectations

In the 2030 Agenda for Sustainable Development, the United Nations and international community established a goal to end hunger by 2030. Currently, the World Health Organization reports that more than 800 million people are facing food shortages worldwise [145]. In addition, the increasing global population is increasing the demand for quality food; therefore, food and cash crops could improve overall crop production.

Figure 7 represents the future challenges agriculture is anticipated to face in 2050. This illustration offers three major problems: (1) feeding 10 billion people, (2) limitations in the expansion of land, and (3) the reduction of greenhouse gases emissions. These challenges lead to new thinking about water scarcity, shrinking arable land, rural labor, climate conditions, and much more. The diminishment in rural populations due to urbanization is not only shrinking communities, but is also leading to ageing populations; therefore, younger growers must step forward to take responsibility. The generation shift and population imbalance create further implications for the workforce and production.

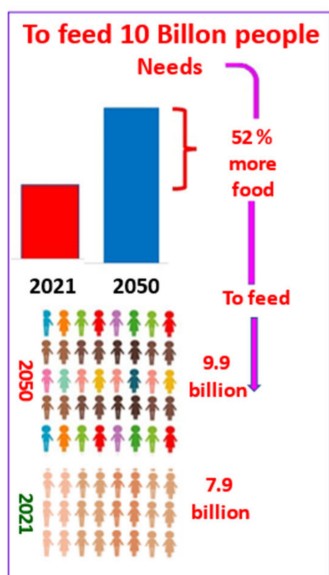 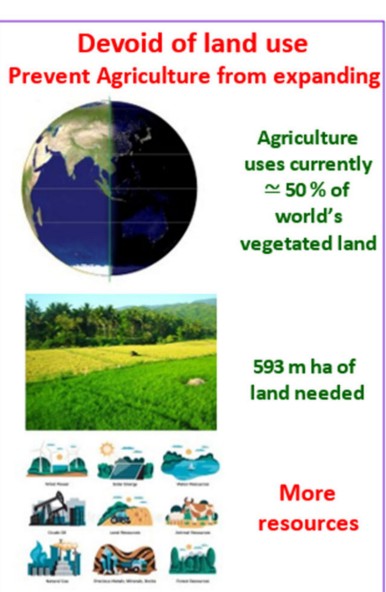 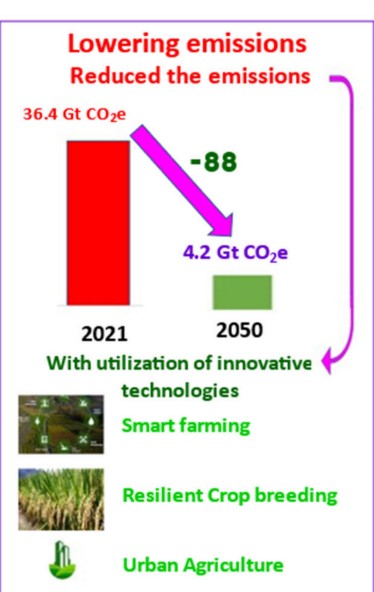

**Figure 7.** Challenges in sustainable future agriculture.

The further shrinking of arable land and the suitability of particular crops in specific regions are due to geographic and ecological conditions. Abrupt weather changes enhance the intensity of environmental issues, such as drought, groundwater depletion, and soil degradation, affecting crop production. Moreover, traditional agricultural methods have historically met food demands by employing fertilizers and pesticides; however, it increases food production only to a certain level and negligent use of chemical deteriorate the environment. In developing countries, various problems facing the agricultural sector include no suitable crop selection, soil testing, efficient irrigation systems, weather forecasting, animal husbandry, etc. Technological advancements have proved beneficial in developed countries, both quantitively and qualitatively, but, in developing countries, 50% of the population is already engaged in agriculture.

The future of agriculture is expected to interconnect with artificial intelligence and big data services. As a result, the systems will converge into a single unit, where farm machinery and management start from seeding to production forecasting. Few of the key technologies and methods are focused on achieving sustainable future agriculture.

### 9.1. Communication

The achievement of the IoT in agriculture mostly depends on connectivity between devices [146]. Most telecom operators provide connectivity services, but represent a small percentage of smart farming as a whole. Cellular operators offer new services to target growers and enhance market facilities, especially in rural areas. The success of cellular technology is feasible when service providers guarantee its benefits, such as flexibility, portability, and extravagance, of both-way communication at low cost. In developing countries, mobile services and smartphone technology offer a hopeful future for farmers to enhance their crop yields. The low power wide area technology (LPWA) is anticipated to play a major role in smart farming agriculture, due to its improved facilities, efficient coverage, low power consumption, and cost economics. The cellular operators with robust IoT create significant returns by offering smart agriculture facilities in collaboration with LPWA technology.

### 9.2. Wireless Sensors and IoT

Placing wireless sensors around the field provides timely information on a real-time basis to farmers in order to make decisions and act in order to obtain higher crop yields. Wireless sensor networks (WSNs) with GPS technology update all information on crop growth and terrain features. Recently, digital images and signal processing offer additional competencies to WSN, and precisely ascertain crop quality and health. The IoT technology can streamline tasks in a predictable manner by diagnosing crop requirements at each stage to maximize their effectiveness. In the future, IoT will be upgraded to the fifth-generation (5G) cellular mobile communication technologies, to provide real-time data to farmers at any time and everywhere. Based on this achievement, around 29 billion IoT-based components are expected to operate in the agriculture sector by the end of 2022. Furthermore, it is expected to create 4.1 million data points daily from farms by 2050 [147].

### 9.3. Drones and Unarmed Vehicles

Farmers widely use drones for crop growth monitoring, spraying nutrient solutions and water, and pesticides in tough terrains and for different crop heights. Drones have proven their value for spraying speed, area coverage, and precision, compared to traditional machinery. Due to advancements in technology, drones are equipped with various sensors, and 3D cameras provide comprehensive capabilities in land management by farmers. With the addition of UAVs in agriculture, many challenges, particularly the incorporation of technologies and use in inclement weather conditions, are addressed by farmers. Other than drones, robotics in agriculture has also enhanced productivity due to higher yields achieved by spraying and weeding without human intervention. The seeding, transplanting, and fruit harvest/picking robots have recently added a new efficiency level to traditional methods.

The UAV technology in smart agriculture provides information on fertilization, irrigation, use of pesticides, plant growth monitoring, weed management, crop disease management, and field-level phenotyping to enhance cultivation practices. A new method of 3D modeling has been used to monitor crop growth parameters to determine the height of maize and sorghum plants under field conditions using UAV, and the average root mean square error (RMSE) of sorghum height with hand sampling field data was 0.33 m [148]. The UAV and 3D models were also restored to extract leaf area index (LAI) in soybean plants, the measured LAI predicted accuracy corresponding to the handheld device ($R^2 = 0.92$) was correlated with destructive LAI measurements ($R^2 = 0.89$) [149].

Weed detection and management were assessed by integrating low-resolution multi-spectral high-resolution RGB images [150] using the Random Forest (RF) technique in field-grown rice and sugar beet crops [151]. Multi-spectral digital images obtained by UAVs are used for evaluating vegetation indices (VIs) and multi-temporal VIs to predict grain yield in wheat [152]. The indices, including the normalized difference vegetation index (NDVI), spectral vegetation index (SVI), and green area index (GAI), are evaluated in wheat crops to predict grain yield [153], monitor breeding [154], detect plant stress caused by yellow rust disease [155], and quantify plant density [156]. The usage of pesticides in agriculture is crucial for crop yields and the environment, and efforts have been made to develop and evaluate an algorithm to self-adjust UAV routes during chemical spraying in a crop field to reduce the waste of pesticides and fertilizers [157].

*9.4. Vertical Farming and Hydroponics*

The shrinking of arable land and rapid urbanization results in greater pressure on the present resources [158], which causes hardships for food production with current agriculture practices. Vertical farming (VF) navigates land and water shortage challenges, and is highly suitable for adoption in nearby cities. Hydroponics plays a key role in lowering water requirements. Hydroponics, along with VF, increases available arable land without distressing forests and other natural habitats. The presence of advanced technologies, especially the IoT, makes the agriculture industry highly remunerative with a reduction in labor requirements and other resources, in addition to minimizing environmental impact.

*9.5. Performance Analysis Using Machine Learning*

Data analytics and machine learning concepts are applied to analyze the real-time data. In crop production, identifying the best genes is an important process that can be conducted using machine learning techniques. In agriculture, machine learning is used to envisage the best genes suited for crop production, especially for selecting seed varieties that are highly suitable to specific climate conditions and locations. Machine learning algorithms identify high demand products and currently unavailable products. Recent developments in machine learning and analytics allow farmers to correctly categorize their harvests before it is processes and delivered to customers.

Machine learning (ML) in big data systems solves the issues related to farmers' decision-making, crops, animal research, land, food availability and security, weather and climate change, and weeds [159]. ML-based applications accommodate a large number of agricultural activities, such as yield prediction based on a deep memory model for maize [160], binary classification model with logistic regression technique to assess rainfall intensity [161], and a short-term memory model to predict soil water content with data parameters of rainfall, temperature, water diversion, evaporation, and time for the next 1, 2, and 7 days with greater $R^2$ compared to artificial neural networks [162]. As a result, the agricultural sector is increasing farmers' incomes, and so communities are further integrated into the agricultural value chain to reduce poverty and provide access to health care, education, and nutritious food for their families [163].

*9.6. Renewable Energy, Microgrids and Smart Grids*

Smart farming requires extensive energy due to power consumption by long-standing sensor placement, use of GPS, and data transmission. Traditionally, using renewable energy sources in remote areas solves long-term power issues. Smart grids and microgrids are integrated into distributed energy sources (DERs). Recent advances in storage devices combine electricity and heat systems to stock energy and use the heat produced.

Globally, smart grid technology enables a smooth transition from traditional to smart energy systems, ensuring energy security. In developing countries, power-strengthening systems integrated with renewable sources have enhanced the transport sector, and increased bioenergy use in the power sector through profuse renewable energy sources identified using smart technologies, such as, energy storage devices, smart appliances,

computational intelligence, and the IoT. For example, the smart grid provides a broad range of opportunities for power sector reform in Nepal, alleviating the rural electricity problem by implementing smart microgrids, and subsequently, connecting to the national grid [164]. The Dayalbagh renewable energy smart microgrid in India is a small-scale electricity system comprising distributed loads and renewable energy resources, acting as a single controllable entity in the grid. The smart microgrids are integrated into renewable resources and form building blocks of smart grids, especially for the dairy plant to produce various dairy products [165].

The mixed integer linear programme (MILP) systematically and efficiently managed energy consumption and subsequently lowered the cost, especially in residential areas, by scheduling the use of smart appliances and charging/discharging electric vehicles (EVs). The model generates its own energy from a microgrid containing solar panels and wind turbines, and forecasts wind speed and solar radiation for effective energy management. MILP-based energy planning sustains the effectiveness and productiveness of energy-efficient techniques [166].

## 10. Conclusions

Smarter and more efficient crop production methodologies are needed to address the issues of shrinking arable land and the food demands of an increasing world population. There is a necessity for everyone to be aware of food security in terms of sustainable agriculture. The growth of new technologies for increasing crop yield and encouraging the adoption of farming by innovative young people as a legitimate profession. This paper emphasized the role of many technologies used for farming, particularly the IoT, in making agriculture smarter and more effective in meeting future requirements. The current challenges faced by the industry and future prospects are noted to guide scholars and engineers. Hence, every piece of farmland is important to enhance crop production by dealing with every inch of land using sustainable IoT-based sensors and communication technologies.

**Author Contributions:** Conceptualization, M.D.; investigation, P.C., K.R.; methodology, M.D., S.P.; resources, K.R., R.K.; supervision, S.P.; visualization, R.K.; writing—original draft, M.D.; writing—review & editing, P.C.; funding acquisition, S.P., validation, R.K. All authors have read and agreed to the published version of the manuscript.

**Funding:** This research and APC were funded by GIZ, Germany by Deutsche Gesellschaft für Internationale Zusammenarbeit (Grant number 81278637).

**Conflicts of Interest:** The authors declare that no competing financial interests or personal relationships could have appeared to influence the work reported in this paper.

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
