# Peer review of "Smart Farming: Internet of Things (IoT)-Based Sustainable Agriculture"

_agriculture, doi:10.3390/agriculture12101745_

Round 1

Reviewer 1 Report

This work proposes an extensive review on IoT based sustainable agriculture. As such, the topic is of interest, however, the paper need some grammar modification.

1. In line 373, “for crop breeding in earlier periods however”, there should be a “,” before however.

In line 375, “In this condition, [70] reported that. Plant pheno”, it should delete the “. ” after “that”.

2. In line 572, there should be a “.” at the end of the sentence. 

once the above concerns are corrected, the manuscript could be accepted.

Author Response

"Please see the attachment

Reviewer 2 Report

(1) The key technologies are not well introduced. It is just simple introduction.  

(2) Some important progress in smart farming needs to be added.

(3) There are too many chapters, and the key points are not highlighted.

(4) The current research progress is not well summarized, and it is recommended to show the current research progress by combining tables and figures.

(5) The explanation is not deep enough, i.e. the Engineer's role in smart farming.

Author Response

"Please see the attachment

Reviewer 3 Report

Manuscript, entitled „Smart Farming:  Internet of Things (IoT) based sustainable Agriculture  . The article reports This paper accentuates the tools, equipment and possibility of wireless sensors application in  IoT agriculture and anticipated challenges faced while relating technology with conventional farming activities. Further, this technical knowledge serves the growers for entire crop periods from sowing to harvest, packing and transport are explicated, there are several shortcomings and there was lack in discuss in several section with  previous  studies that should be included in order to enhance the final manuscript for the readers

Line 29.please add the citation at the end of the sentence.

Lines 31.please add the citation at the end of the sentence.

Lines 39.please add the citation at the end of the sentence.

Line 43.please add the citation at the end of the sentence.

Line 54.please add the citation at the end of the sentence.

Line 61.please add the citation at the end of the sentence.

Figure 3. Did you establish this figure or you take from other paper? If that, please add the citation for this figure.

Line 102.please add the citation at the end of the sentence.

Line 104.please add the citation at the end of the sentence.

Line 114.please add the citation at the end of the sentence.

Line 140.please add the citation at the end of the sentence.

Line 162.please add the citation at the end of the sentence.

Line 180. 4.2. Sensor technologies. This section need to add more information about Sensor technologies with more citations.

Line  185. 4.3. Variable-Rate of technology (VRT) and Grid soil sampling . The authors should present the previous work in this area. There is no previous work was discussed.  

Line 196.please add the citation at the end of the sentence.

Line  202. 4.5. Crop management. The authors should present the previous work in this area. There is no previous work was discussed.  

Line  202. 4.6. Soil and plant sensors. The authors should present the previous work in this area. There is no previous work was discussed.  

Line 211.please add the citation at the end of the sentence.

Line 221.please add the citation at the end of the sentence.

Line  202. 4.6. Soil and plant sensors. The authors should present the previous work in this area. There is no previous work was discussed.  

Line  202. 4.9. Yield monitor. The authors should present the previous work in this area. There is no previous work was discussed.  

Line 234.please add the citation at the end of the sentence.

Line  235. 5. Applications in Agriculture. The authors should present the previous work in this area. There is no previous work was discussed.

Figure 5. Did you establish this figure or you take from other paper? If that, please add the citation for this figure.

Line  245. 5.1. Soil mapping and Plant Monitoring. The authors should present the previous work in this area. There is no previous work was discussed.

Line  334. 6.1. Greenhouse farming and protected cultivation. The authors should present the previous work in this area. There is no previous work was discussed.

Line  348. 6.2. Hydroponics.  The authors should present the previous work in this area. There is no previous work was discussed.

Line  523. 9.3. Drones and unarmed vehicles.  The authors should present the previous work in this area. There is no previous work was discussed.

Line  545. Performance analysis using Machine learning.  The authors should present the previous work in this area. There is no previous work was discussed.

Line  545. 9.6. Renewable energy, Microgrids, and Smart grids.  The authors should present the previous work in this area. There is no previous work was discussed.

Author Response

"Please see the attachment

Round 2

Reviewer 2 Report

(1) There are too many chapters. The author should summarize the manuscript.

(2) The reference number in this manuscript should be in order.

Author Response

The corrections indicated by the reviewer are carried out. 

Reviewer 3 Report

The authors  have improved the manuscript. But citations must be arranged. for example, the authors put the citation number 83 after citation 1.

Author Response

The references are arranged as per the suggestion
